# Chromosome-Length Assembly of the Baikal Seal (*Pusa sibirica*) Genome Reveals a Historically Large Population Prior to Isolation in Lake Baikal

**DOI:** 10.3390/genes14030619

**Published:** 2023-02-28

**Authors:** Aliya Yakupova, Andrey Tomarovsky, Azamat Totikov, Violetta Beklemisheva, Maria Logacheva, Polina L. Perelman, Aleksey Komissarov, Pavel Dobrynin, Ksenia Krasheninnikova, Gaik Tamazian, Natalia A. Serdyukova, Mike Rayko, Tatiana Bulyonkova, Nikolay Cherkasov, Vladimir Pylev, Vladimir Peterfeld, Aleksey Penin, Elena Balanovska, Alla Lapidus, Stephen J. OBrien, Alexander Graphodatsky, Klaus-Peter Koepfli, Sergei Kliver

**Affiliations:** 1Computer Technologies Laboratory, ITMO University, 19701 Saint Petersburg, Russia; 2Department of Natural Sciences, Novosibirsk State University, 630090 Novosibirsk, Russia; 3Department of the Diversity and Evolution of Genomes, Institute of Molecular and Cellular Biology SB RAS, 630090 Novosibirsk, Russia; 4Skolkovo Institute of Science and Technology, 121205 Moscow, Russia; 5Applied Genomics Laboratory, SCAMT Institute, ITMO University, 9 Ulitsa Lomonosova, 191002 Saint Petersburg, Russia; 6Human Genetics Laboratory, Vavilov Institute of General Genetics RAS, 119991 Moscow, Russia; 7Tree of Life, Wellcome Sanger Institute, Cambridge CB10 1SA, UK; 8Centre for Computational Biology, Peter the Great Saint Petersburg Polytechnic University, 195251 St. Petersburg, Russia; 9Center for Bioinformatics and Algorithmic Biotechnology, St. Petersburg State University, 199034 St. Petersburg, Russia; 10Laboratory of Mixed Computations, A.P. Ershov Institute of Informatics Systems SB RAS, 630090 Novosibirsk, Russia; 11Laboratory of Human Population Genetics, Research Centre for Medical Genetics, 115522 Moscow, Russia; 12Baikal Branch of State Research and Industrial Center of Fisheries, 670034 Ulan-Ude, Russia; 13Institute for Information Transmission Problems of the Russian Academy of Sciences, 127051 Moscow, Russia; 14The Center for Genome Architecture, Department of Molecular and Human Genetics, Baylor College of Medicine, Houston, TX 77030, USA; 15Guy Harvey Oceanographic Center, Halmos College of Arts and Sciences, NOVA Southeastern University, Fort Lauderdale, FL 33004, USA; 16Smithsonian-Mason School of Conservation, George Mason University, 1500 Remount Road, Front Royal, VA 22630, USA; 17Center for Species Survival, Smithsonian’s National Zoo and Conservation Biology Institute, 1500 Remount Road, Front Royal, VA 22630, USA; 18Center for Evolutionary Hologenomics, The Globe Institute, The University of Copenhagen, 5A, Oester Farimagsgade, 1353 Copenhagen, Denmark

**Keywords:** *Pusa sibirica*, conservation, pinnipeds, demography, heterozygosity

## Abstract

*Pusa sibirica*, the Baikal seal, is the only extant, exclusively freshwater, pinniped species. The pending issue is, how and when they reached their current habitat—the rift lake Baikal, more than three thousand kilometers away from the Arctic Ocean. To explore the demographic history and genetic diversity of this species, we generated a de novo chromosome-length assembly, and compared it with three closely related marine pinniped species. Multiple whole genome alignment of the four species compared with their karyotypes showed high conservation of chromosomal features, except for three large inversions on chromosome VI. We found the mean heterozygosity of the studied Baikal seal individuals was relatively low (0.61 SNPs/kbp), but comparable to other analyzed pinniped samples. Demographic reconstruction of seals revealed differing trajectories, yet remarkable variations in Ne occurred during approximately the same time periods. The Baikal seal showed a significantly more severe decline relative to other species. This could be due to the difference in environmental conditions encountered by the earlier populations of Baikal seals, as ice sheets changed during glacial–interglacial cycles. We connect this period to the time of migration to Lake Baikal, which occurred ~3–0.3 Mya, after which the population stabilized, indicating balanced habitat conditions.

## 1. Introduction

The origin of the endemic freshwater Baikal seal (*Pusa sibirica*), and its phylogenetic relationships with other seals, is a long-term biogeographic issue. Lake Baikal is connected to the Kara Sea in the Arctic Ocean through the Angara–Yenisei river system. There is no definitive data of how seals migrated to a landlocked habitat, but there are several hypotheses. The most plausible current view suggests that their migration routes led from the Arctic waters along the Yenisei, or through glacial and ice-dammed lakes, which existed in the Middle Pleistocene during at least four continental glaciation periods in central Siberia [1,2]. Estimates of the migration time, inferred from phylogenetic analyses (primarily mitochondrial DNA), encompass a broad period, depending on the data and methods used: from 2–3 million years ago (Mya) (based on MP, ML, and NJ methods) [2], to 1.5–2 Mya (based on BI) [3], and 0.4 Mya (based on UPGMA and NJ methods) [4]. At the moment, researchers are sure only of the Arctic origin of the Baikal seal [5]. This hypothesis is supported by several Arctic adaptations of the modern pagophilic species: breeding on ice; the white natal fur of pups, providing camouflage on snow-covered ice; the ability to spend considerable time and cover long distances underwater, considering the chance to be trapped under the thick ice; and the instinct to breathe from pockets of exhaled air that accumulates under the ice [6,7,8]. Similar behavior is typical for the Baikal seal’s Arctic relatives—the ringed seal (*Pusa hispida*) and Caspian seal (*Pusa caspica*) [9,10,11].

The earliest records of seals in Lake Baikal appeared due to their interaction with humans. Hunter-fishermen began depicting seals on rocks, and as bone and stone carvings, more than 7000 years ago [7]. Zooarchaeological analyses of seal fossils showed that seal hunting began at least 9000 years ago, in the Middle Holocene [8]. Harvesting seals for their fat and fur began in the 19th and 20th centuries, and later, it continued in order to protect commercially important fish species. Poaching pressure became several times greater than that from legal harvesting, which became heavily regulated following a 1986 act by the Soviet government on the protection of marine mammals, that was caused by a global ban on commercial whaling, including all marine mammals [12]. The establishment of several federal nature protection territories along the Lake Baikal coastline, including the Zabaikalskiy and the Pribaikalskiy nature reserves (zakazniks), further contributed to conservation efforts. It was later revealed that, contrary to popular belief, the basis of the seal’s diet does not include commercially important fish, such as the omul (*Coregonus migratorius*), and does not pose a threat to fisheries [13,14]. Since Baikal seals lack molars for chewing, they prefer medium-sized fish, such as the Baikal oilfish, or golomyanka (genus *Comephorus*), which are swallowed whole. Baikal seals also possess highly specialized, comb-like postcanine teeth, which are used to forage the endemic freshwater amphipod *Macrohectopus branickii* at high rates [15]. Nevertheless, seal hunting continues to this day, mainly targeting molted pups, as the fur of young individuals is considered to be commercially more valuable.

Traditional methods of population size estimation for the Baikal seal remain challenging, due to their cautious nature and a predominantly underwater lifestyle, where they spend 80–90% of their time [5,16,17,18]. According to different sources, the census population size was estimated at 108,200 individuals in 2013, according to the Russian Ministry of Natural Resources [19], 94,600–137,400 in 2018, and 164,500 in 2021, according to the Baikal branch of the Russian Federal Research Institute of Fisheries and Oceanography (“VNIRO”), “BaikalNIRO” [20]. Estimates based on breeding snow dens are also complicated, as dens are well fortified and camouflaged. Dens have blowholes opening into the water, that are used by females to feed their pups, and are also used by both to escape, in case of danger [7,8]. Seals can be seen above the water surface in the spring, when they use the ice floes to complete their molt and migrate north across the lake. They sometimes lay on rocks near the shore, but only in the spring and summer. However, such individuals usually include only a small percentage of the population (at different times 0.1–10%), often representing tired or sick individuals or those that were not able to complete molt, due to an early ice break [21]. There are many locations suitable for shore haulouts. However, due to negative anthropogenic impact, most of those locations are avoided by seals, which should be regarded as a loss of potential habitat [21].

Climatic changes, and consequent habitat loss or increase, can affect closely related species, especially if they have overlapping areas or occupy similar ecological niches. Baikal seals are now isolated in the lake, but their ancestors were marine animals [3]. Therefore, it makes sense to compare both the modern population size and demographic history of the Baikal seal with other species from the *Phoca*-*Pusa*-*Halichoerus* lineage. Currently, whole genome data is available for at least the gray seal (*Halichoerus grypus*), spotted seal (*Phoca largha*), and harbor seal (*Phoca vitulina*). Populations of these species mostly inhabit northern seas and ocean regions (Figure 1). The western Atlantic gray seals are believed to have split from the rest of the population of gray seals at ~0.03 Mya [22]. An expansion of their populations, caused by habitat increase, is believed to have occurred after the last glacial maximum (LGM), at ~0.02 Mya [22]. But in the late 1800s, the eastern Canadian population included only a few dozen individuals, due to harvesting for oil [23]. Nowadays, the numbers have increased to around 450,000 individuals [24]. Since the western Atlantic population went through periods of severe downsizing, and we can exclude trans-Atlantic movement, the modern population must have originated from small remaining populations, and recolonized the coasts of the USA and Canada [25]. The information on spotted seals is rather poor. They are well known to have been harvested for centuries in Russian and Chinese waters, which had a great impact on population size [26]. Since the 1980s, the Bering population has grown from ~248,000 individuals to ~460,000 [27]. However, the southern population is significantly lower, including ~3300 individuals [26]. An interesting feature of gray seals and spotted seals, is their ability to cover huge distances in seasonal feeding, which could lead to genetic exchange between breeding populations and colonies [28,29,30].

Only a few genetic, and no genomic, studies on Baikal seals have been published. Most of them are based on mtDNA only [2,3,4,31,32]. However, mtDNA is very short and provides a relatively small number of informative markers. The current trend is to use whole genome data, and chromosome-length assemblies, even in studies of nonmodel organisms [33,34,35,36]. Our study fills this gap in the genomics of pinnipeds. We generated the first chromosome-length assembly of the Baikal seal genome. This allowed us to elucidate the genetic diversity, demographic history, and phylogenomics of the Baikal seal, and to compare its features with the additional pinniped species mentioned above.

## 2. Materials and Methods

### 2.1. Sequencing of Baikal Seal Individuals

For whole genome sequencing, we used primary fibroblast cell lines of one male (PUSI1m) and one female (PUSI1f) Baikal seal, obtained from the Novosibirsk Cell Line Collection, located at the Institute of Molecular and Cell Biology, Siberian Branch, Russian Academy of Sciences. Sample collection, transportation, and cell line establishment were previously described in detail in [37].

The male individual was chosen to generate a de novo Baikal seal genome assembly. For this sample, we created one paired-end, and multiple mate pair (jumping), libraries, of different insert sizes, and sequenced these on an Illumina HiSeq 2000 or 2500 (Illumina, Inc., San Diego, CA, USA) instrument. We also generated and sequenced one Hi-C library on an Illumina NovaSeq6000 (Illumina, Inc., San Diego, CA, USA) instrument, which was used to scaffold the assembly to chromosome length. A detailed description of the libraries is provided in Appendix A. For the female individual, we generated and sequenced only one paired-end library.

### 2.2. Filtration, QC, and Postprocessing of Sequencing Data

Quality control of raw and filtered reads was performed using FastQC v0.11.9 [38]. Distributions of 23-mers before and after read filtration were counted using Jellyfish v2.2.10 [39], with parameters ‘-m 23 -s 30G -t 24′ for *jellyfish count*, and ‘-t 4 -l 1 -h 1,000,000 -i 1′ for *jellyfish histo*, and then visualized using KrATER v2.5 (https://github.com/mahajrod/krater (accessed on 21 December 2022)), with parameters ‘-m 23 -w 8 -g 150 -u 1′ to check for possible anomalies and signs of contamination. Trimming of Illumina adapters and quality filtering were performed using Trimmomatic v0.39 [40], with parameters ‘ILLUMINACLIP:$adapter:2:30:10 SLIDINGWINDOW:4:15 MINLEN:5′. Baikal seal mate pair libraries were additionally processed using NextClip [41], with default parameters, to remove the Nextera adapter and to extract true mate pair reads. To achieve equal coverage and maximize similarity among samples of all seal species, we cut all reads to 100 bp (the read length used to sequence the Baikal seal samples), and downsampled them to 20× coverage. For downsampling we used FaCut (https://github.com/mahajrod/Facut (accessed on 21 December 2022)).

### 2.3. Assembly of Baikal Seal Reference Genome

A chromosome-length genome assembly of the Baikal seal was generated in three stages. First, we generated a draft assembly from filtered paired-end and mate pair libraries of the reference individual, with the Plantanus v1 genome assembler, using default parameters [42]. Next, we scaffolded the draft assembly to chromosome level, using the reads from the Hi-C library and the 3D-DNA pipeline, with default parameters [43]. Finally, the generated scaffolds were manually verified and corrected using Juicebox [44].

### 2.4. Genomes of other Pinnipeds and QC of All Assemblies

For comparative analysis, we used chromosome-length assemblies and sequencing data of three related pinniped species [45,46,47,48]: the gray seal *(* SRX12373285), spotted seal (SRX9606535, SRX3524687), and harbor seal. Assemblies and reads were downloaded from the DNA Zoo (https://www.dnazoo.org/ (accessed on 21 December 2022)) and NCBI Sequence Read Archive (SRA) (Table 1). Genome assemblies of all species were assessed for general quality metrics and gene completeness using BUSCO v5 and the mammalia_odb10 and carnivora_odb10 databases [49].

### 2.5. Whole Genome Alignment and Connection between Assembly and Karyotype

First, we softmasked tandem and interspersed repeats in the genomes of the four pinniped species and the stone marten genome (*Martes foina*), using Tandem Repeats Finder v4.09.1 [50], WindowMasker 1.0.0 [51], and RepeatMasker v4.1.2.p1 [52]. Second, a whole genome alignment of the five analyzed species was performed, using Progressive Cactus [53], with default parameters. Next, synteny blocks were extracted from the multiple whole genome alignment using halSynteny v2.2 [54], with the options—minBlockSize 50000—maxAnchorDistance 50000. Finally, we visualized synteny blocks using ChromoDoter (https://github.com/mahajrod/ChromoDoter (accessed on 21 December 2022)) and the draw_synteny.py script from the MACE package (https://github.com/mahajrod/mace (accessed on 21 December 2022)). Dot plots, generated by ChromoDoter, were manually compared to Zoo-FISH experiments previously published for PSIB, HGRY, and PLAR karyotypes, with MFO chromosome libraries used as probes [37,55,56,57,58].

### 2.6. Read Alignment, Coverage Estimation, and Variant Calling

Filtered paired-end reads of analyzed individuals were aligned to the corresponding reference genome assemblies with BWA v0.7.17, using the MEM algorithm [59], followed by duplicate marking, sorting, and indexing using Samtools v2.30.0 [60]. For all generated alignments we calculated per-base coverage using Mosdepth v0.3.1 [61], and visualized it on heatmaps using the MACE package (https://github.com/mahajrod/MACE (accessed on 21 December 2022)).

To set the correct ploidy for the X chromosome during variant calling, we identified the pseudoautosomal region (PAR). First, median coverage was determined in stacking windows of 10 kbp. Adjacent windows with median coverage ≥ 70% of the whole genome level were merged. Then all the combined windows with the median coverage of the intermediate gap ≥ 70% of the whole genome level were merged. This procedure improves the accuracy of detecting the PAR coordinates.

SNPs and short indels were detected using bcftools v1.15 [60]. First, we used the *bcftools mpileup* command, with default parameters. Next, the *bcftools call* command was run, with the set PAR coordinates. Finally, low-quality variants were removed with the bcftools *filter* command, with the following parameters: “QUAL < 20.0||(FORMAT/SP > 60.0|FORMAT/DP < 5.0|FORMAT/GQ < 20.0)”. Finally, only variants with coverage in the 30–250% range, relative to the mean whole genome coverage, were retained.

### 2.7. Heterozygosity

For analysis of heterozygosity, we applied a sliding window approach. Heterozygous SNPs were counted in overlapping sliding windows of 1 Mbp, with a step size of 100 kbp, and counts were scaled to SNPs per kbp. Heterozygosity counts were visualized on chromosome scaffolds and violin plots, using the MACE package and Matplotlib library (https://matplotlib.org (accessed on 21 December 2022)).

### 2.8. Demographic Reconstruction

Demographic history inference was performed using PSMC v0.6.5 [62], with the following parameters: -N25 -t15 -r5 -b -p ‘4 + 25 × 2 + 4+6′. We used mutation rate and generation time values based on scientific publications of the target species, or from closely related species [22,63,64,65,66,67]. For mutation rate, we used values of 0.7 × 10^−8^, 1.2 × 10^−8^, and 2.5 × 10^−8^ substitutions per site per generation, which is in the range of commonly used values for mammals [68,69,70]. Generation times were chosen to be equal to 21.6 years for the Baikal seal [19], 16.5 years for gray seal [71], and for spotted seals we used 14.8 years, which is the same as commonly used for its close relative, the harbor seal [72,73]. Effective population size trajectories for each species were visualized with the *psmc_plot* command.

### 2.9. Phylogenetic Tree Reconstruction and Dating

We reconstructed a phylogenetic tree using multiple alignments of coding sequences of single-copy orthologous genes (BUSCOs). In the reconstruction, we included 11 species: cheetah (*Acinonyx jubatus*), mountain lion (*Puma concolor*), domestic dog (*Canis lupus familiaris*), brown bear (*Ursus arctos*), red panda (*Ailurus fulgens*), walrus (*Odobenus rosmarus*), bearded seal (*Erignathus barbatus*), harbor seal (*Phoca vitulina*), spotted seal (*Phoca largha*), gray seal (*Halichoerus grypus*), and Baikal seal (*Pusa sibirica*). Corresponding assemblies were downloaded from the NCBI Genome database (https://www.ncbi.nlm.nih.gov/genome/ (accessed on 21 December 2022)) and DNA Zoo (https://www.dnazoo.org/ (accessed on 21 December 2022)): AJUB, ID:aciJub1_HiC [74]; PCON, ID:PumCon1.0_HiC [75]; CLUP, ID:canFamDis_HiC [76]; UARC, ID:ASM358476v1_HiC [77]; AFUL, ID:ASM200746v1_HiC [78]; OROS, ID:Oros_1.0_HiC [79]; EBAR, ID:Erignathus_barbatus_HiC [80]; PVIT, ID:GSC_HSeal_1.0_HiC [81]; PLAR, ID:Phoca_largha_HiC [80]; HGRY, ID:Halichoerus_grypus_HiC [43,44,80]. BUSCO sequences were generated using BUSCO v.5.4.2 [49] with the Mammalia_odb v.10 (2021-02-19) database of orthologs (9226 BUSCOs). Only single-copy sequences common for all species were included in the analysis. Multiple codon alignments were performed separately for each ortholog using PRANK v.170427 [82], followed by filtration of the hypervariable and poorly aligned regions using GBlocks v.0.91b [83]. A phylogenetic tree, based on the concatenated alignments of 5025 BUSCOs (8,153,886 Mbp), was estimated using the maximum likelihood method implemented in the IQ-Tree v.2.2.0 [84], with automatic selection of the best-fitting DNA substitution model using ModelFinder [85]. For the evaluation of node support, 1000 bootstrap replicates were generated.

Dating of the phylogenetic tree was performed using the MCMCtree tool from the PAML package v4.7 [86], with the HKY85+G model of nucleotide substitution and 220,000 MCMC generations, of which the first 20,000 generations were discarded as burn-in. Divergence times were assessed using 4-fold degenerate sites, which were extracted from the concatenated codon alignment generated during the tree reconstruction stage. Fossil calibrations used for dating are listed in Appendix A. The final dated tree was visualized using FigTree v.1.4.4 (http://tree.bio.ed.ac.uk/software/figtree (accessed on 21 December 2022)).

## 3. Results

### 3.1. Genome Size and Assembly, Repeat Content

We assembled the first chromosome-length reference genome of a male Baikal seal (PSIB) individual using 2386.3 million paired-end reads, 735.1 million mate pair reads of different insert sizes, and 1348.5 million Hi-C reads (Appendix A). An estimation based on the distribution of 23-mers extracted from filtered paired-end reads, showed a genome size of 2.12 Gbp for the male individual, and 2.22 Gbp for the female (Appendix A). Downsampling to the standardized 20× coverage did not change the estimates (Appendix A). Two closely related species, the gray seal and spotted seal, for which raw sequencing data were available, demonstrated similar values: 2.29 Gbp and 2.37 Gbp, respectively. The Baikal seal assembly had a total length of 2.35 Gbp, which is slightly higher than the genome size estimate. Scaffold N50 reached 151.5 Mbp (Table 1), and we observed a dramatic 29-fold difference in length between the 16th (57.63 Mbp) and 17th (1.99 Mbp) longest scaffolds. Thus, our assembly includes 16 chromosomal scaffolds (C-scaffolds), which exactly fit the number of chromosome pairs in the karyotype (2n = 32). A BUSCO-based assessment indicated high completeness of the assembly. For the mammalia_odb10 (9226 BUSCOs) and carnivora_odb10 (14,502 BUSCOs) databases, we found 86.5% and 88.0% complete and single-copy, and 1.5% and 1.6% complete and duplicated BUSCOs, respectively. These, and other metrics for previously published pinniped species, are somewhat higher, yet comparable (Table 2, Appendix A). Interspersed repeat content among pinniped species is also similar, with LINEs, SINEs, and LTRs encompassing 3.16%, 19.18%, and 4.92%, respectively, of the Baikal seal genome (Appendix A).

### 3.2. Connection between Assembly and Karyotype

Of the four analyzed species, Zoo-FISH data is available only for the Baikal seal and harbor seal, making it possible to assign C-scaffolds to particular chromosomes in the karyotype for these two species. Following identification, C-scaffolds were assigned to the corresponding chromosome numbers with the prefix “chr”. For the two other species (HGRY and PLAR), such an assignment was possible only for the X chromosome, because it is highly conserved across Carnivora and mammals [87]. The remaining C-scaffolds were numbered from longest to shortest, with the prefix “aut” indicating “autosomal” (Appendix A). For the Baikal seal and spotted seal, we found no contradictions between the Zoo-FISH data and whole genome alignments (WGA).

Synteny blocks (>50 kbp in length) extracted from the whole genome alignment, showed a high similarity between PSIB, HGRY, PLAR, and PVIT (Figure 2). Species with assigned chromosomes (Baikal seal and harbor seal) have precise correspondence between chromosome ids, i.e., chr1 of one species corresponds exactly to chr1 of the second. However, C-scaffolds of several chromosomes (chr2, chr4, chr3, chr7, chr10, chr13, chr14, chr15, and chrX) are inverted in the assemblies. By ordering autosome C-scaffolds by length in the spotted seal (PLAR) and gray seal (HGRY), we found them to be slightly rearranged: aut1 and aut2 correspond to chr2 and chr1 of the other species, and aut13 and aut14 correspond to chr14 and chr13, respectively.

We detected dozens of short inversions (Appendix A) and only three relatively large inversions of megabase-scale on chr 6. One of them (Figure 2, enlarged pane, yellow) is common to the harbor seal (PVIT) and spotted seal (PLAR), and is probably a synapomorphy of the genus *Phoca*, which includes only these species. The second and third inversions (Figure 2, enlarged pane, orange) are unique to the harbor seal (PVIT).

### 3.3. Heterozygosity and SNP Density

Using a coverage-based approach, we verified the morphological sex of the studied individuals, and detected coordinates of the pseudoautosomal region (PAR) in the male Baikal seal and male gray seal. In both species, the PAR encompasses slightly different lengths at the ends of the X chromosomes: 6.75 Mbp in the Baikal seal and 5.88 Mbp in the gray seal (Appendix A). Heterozygous SNPs, counted in sliding windows of 1 Mbp, with a 100 kbp step size, showed a median heterozygosity (hetSNPs per kbp) of 0.48 for the gray seal, 0.67 and 0.62 for spotted seals, and 0.61 and 0.66 for male and female Baikal seals, respectively (Figure 3F). Mean values did not differ significantly from the median ones (Table 3). Exclusion of the X chromosome does not show any remarkable changes in values: 0.5 for gray seals, 0.68 and 0.63 for spotted seals, and 0.62 and 0.67 for male and female Baikal seals, respectively (Appendix A).

Heatmaps of SNP density distribution showed a slightly higher number of heterozygous SNPs on the ends of chromosomes in all individuals, and relatively short regions (dark blue) of low heterozygosity in the middle of several chromosomes (Figure 3). The PAR region had SNP densities comparable to the autosomes in both males and females. Overall, however, SNP densities were generally similar among the three seal species we compared.

### 3.4. Demographic Reconstruction

A PSMC-based reconstruction of the historical demography for the Baikal seal, gray seal, and spotted seal resulted in different trajectories of effective population size (*Ne*) changes, but individuals from the same species showed very similar tracks (Figure 4). THe gray seal, spotted seal, and Baikal seal share one significant Ne peak in the time intervals ~1–0.8/1.5–2/3–4 Mya (here and below we provide dating for three mutation rates in the following order: 2.5 × 10^−8^/1.2 × 10^−8^/0.7 × 10^−8^). They also share a second peak that is more extended over time, and varies in intensity among the three species: for the gray seal it is higher, while for the spotted seal it is lower, with both occurring at ~0.2/0.4/0.7 Mya with respect to μ. At the bottom of the first population decline, ~0.3/0.6/1 Mya, Ne decreased by ~40% in the gray seal and ~75% in the spotted seal. For Baikal seals, at ~0.8/1.8/3 Mya, the most dramatic decline had started, with a roughly 10-fold decrease in Ne, according to the demographic reconstruction. After the second, relatively small decrease, the spotted seal population stabilized at ~0.1/0.2/0.4 Mya, and even showed a trend for growth, but the Ne for the gray seal continued to decrease until modern times. The decline in population size for the Baikal seal was followed by a small growth in Ne at ~0.25/0.5/0.9 Mya, followed by stabilization at that level.

### 3.5. Phylogenomics and Molecular Dating

We reconstructed and dated a maximum likelihood phylogenetic tree of five phocid seals and six other carnivore species, using five fossil calibrations (Appendix A). Two felid species (cheetah and jaguarundi) were used as the outgroup to root the tree and to provide a fossil-dated root. All nodes on the tree were supported at 100% based on 1000 bootstrap replicates. According to our results, HGRY and PSIB are sister species, whereas *Phoca* (PLAR, and PVIT) form an outside group (Figure 5). The divergence between HGRY and PSIB was assessed to occur around ~1.6 Mya, with a 95% confidence interval (CI) encompassing 2.2–1.1 Mya. For PLAR and PVIT, speciation occurred later, at ~0.7 Mya (95% CI 1–0.4 Mya). The division between *Phoca* and *Halichoerus-Pusa* lineages was dated to ~2 Mya, with a 95% CI of 2.6–1.4 Mya.

## 4. Discussion

### 4.1. Chromosome-Level Assemblies and Karyotypes

Assignment of chromosome numbers to C-scaffolds can only be established if Zoo-FISH or other marker-based mapping data is available [88]. Such data were previously published for the Baikal seal and harbor seal. Our findings show that chr1 in the assemblies (specifically, the C-scaffold) of both species is shorter than chr2. However, in the karyotypes, chr1 is longer than chr2. Such a contradiction may be explained by the presence of an unassembled heterochromatin region in the short arm of chr1, between segments homologous to parts of dog chr5 and chr9 [37]. This region is clearly visible using both C-and G-banding methods, and was identified to be associated with the nucleolus organizer region (NOR) [56]. The NOR consists of multiple ribosomal DNA (rDNA) copies and is impossible to assemble at full length from short reads. Usually, only a few copies of rDNA monomer fragments are present in genome assemblies. Manipulations of reads and contigs, and even additional sequencing (preferably using long read sequencing), are required to generate the correct sequence of the rDNA monomer [89].

Other chromosomes possibly not ordered by length, include chr13 and chr14. The length of these chromosomes seems to be similar, both in the assemblies and karyotypes. The most likely explanation for the discordance is the limited precision for estimation of chromosome length. For the gray seal and spotted seal, we lack Zoo-FISH data. To emphasize this difference, and issues with chr1, chr2, chr13, and chr14, we used the prefix “aut” (instead of “chr”) for C-scaffolds of the two latter species, and assigned numbers to them according to their length (Figure 2 and Figure 3), using the standard rules of chromosome nomenclature [90,91].

### 4.2. Genome Rearrangements and Speciation

The genus *Phoca* contains only two species, the harbor seal (PVIT) and spotted seal (PLAR), both of which were included in our analysis. We detected a single four megabase synapomorphic inversion for this genus, and two more inversions of the same scale specific for the harbor seal. All three inversions are located nearby to one another, on chromosome 6. These findings immediately raise the question of whether these genome rearrangements were associated with divergence of the genus *Phoca,* and speciation within it. Inversions are well known to partially suppress crossover events [92] within themselves, due to unsustainable meiotic products. This leads to linkage disequilibrium [93] and reduced fertility of individuals with heterozygous inversions, and might lead to partial reproductive isolation [94,95,96]. Multiple or nested inversions may aggravate the effect [97]. The second result of such rearrangements, is a significant change in the genome environment, which might affect expression [98], with multiple possible consequences in phenotype or behavior, depending on what genes are included in the inversion. The spotted seal, harbor seal, and gray seal have partly overlapping ranges (Figure 1), and reproductive isolation based on genetic background might shape the speciation within the Phocidae [99]. On the other hand, the detected inversions might be just a genetic feature of the specific individuals that were sequenced. This is less likely for the *Phoca*-specific inversion, as it was detected in two different species, but for the two other inversions detected in the harbor seal, additional individuals representing different populations should be sequenced to further explore the frequency of these inversions.

### 4.3. Heterozygosity

Heterozygosity in pinniped species is known to be low [22,26,100]. According to our results, the Baikal seal, the only pinniped among those analyzed that occurs within a landlocked habitat, showed a mean heterozygosity of 0.63 and 0.68 SNP/kbp for male and female individuals, respectively, comparable to that found in marine pinniped species. Moreover, it was even higher than for gray seals (0.49 SNP/kbp) (Figure 2A,B,E). Low heterozygosity is often associated with reduced fertility and survival rate, and might be a forerunner of further decline, or even extinction [101]. However, nowadays the population trend of Baikal seals is recognized as stable, and the species has the status of “least concern” (LC) on the IUCN Red List of Threatened Species [19]. By comparing heterozygosity levels of representative species from the *Felidae*, *Ailuridae*, *Mustelidae*, and *Phocidae*, we observe that pinnipeds are the only species to have a low heterozygosity level without being listed within a threatened status category (Figure 6).

Reduced heterozygosity can be a sign of extensive inbreeding in an enclosed ecosystem. Baikal seals are effectively trapped in the lake, but Lake Baikal occupies a vast area (with a surface area of 31,722 km^2^), with a very diverse and endemic flora and fauna. Current census count estimates vary between 82,500 and 115,000 seals, which demonstrates the sufficient capacity and resources of the lake to support a large population of a large-bodied predator such as the Baikal seal [15,19]. For comparison, the population of ringed seals, whose area encompasses the entire Arctic Ocean and northern parts of the Atlantic and Pacific Oceans, was estimated to reach “only” ~1.5 million individuals [73]. Another possible explanation to consider for the reduced heterozygosity, is that Baikal seals could have experienced a decline in their initial population size, and hence their genetic diversity, during their long migration to, and isolation in, Lake Baikal. The relatively uniform distribution of heterozygous sites across the genome (Figure 2A,B) may indicate that the current population descended from only a small group of animals, that separated from their ancestors and settled the current habitat [100].

### 4.4. Interpretation of Demographic History Reconstruction

The timescale for PSMC-based reconstruction of demographic history is dependent on the ratio of two parameters: generation time and mutation rate (g/μ). Neither parameter is known with certainty for the Baikal seal. We used a mutation rate (μ) of 2.5 × 10^−8^ substitutions per site, based on the available data for cetaceans and what is commonly used for mammals, 1.2 × 10^−8^ for medium-sized mammals with approximately the same body size as seals, and 0.7 × 10^−8^ for pinnipeds [69,102]. Smaller mutation rates result in stretching out the timeline and causing changes in Ne to be dated older (Appendix A). For generation times, we used values from the IUCN Red List of Threatened Species, which are available for all our species except spotted seals. Under this standard, generation length is defined as “the average age of parents of the current cohort, reflecting the turnover rate of breeding individuals in a population” [103].

The strong connection of Baikal seals to ice suggests an Arctic origin of the species [2,104]. According to multilocus and mitogenome-only phylogenies [3,31], the closest relative of the Baikal seal is the ringed seal, which inhabits the Arctic Ocean and several sub-Arctic seas, such as the Baltic Sea. Ringed seals have a similar association with ice, the same structure of snow dens, and can maintain breathing holes through at least 1.8 m (6 feet) of ice [105]. Such facts make it reasonable to speculate about what part of the Baikal seal’s reconstructed demographic trajectory is shared with that of the ringed seal via recent common ancestry. These two species are estimated to have diverged ~1.15 Mya (95% CI: 1.6–0.72 Mya) [31] or ~1.5–1.7 Mya [3]. However, such a hypothesis could be verified only after whole genome sequencing of ringed seal individuals. But glacial–interglacial cycles during the Pleistocene likely had an effect on the demographic history of all seals, including the Baikal seal [99].

The Mid-Pleistocene Transition (MPT) lasted from approximately 1.25 Mya to 0.7 Mya [106,107]. Ocean sediments show that during the MPT, there occurred a shift in terminations, switches from glacial to interglacial (G–IG) climates, from a symmetric 41 ky to an asymmetric 100 ky periodicity [106,107]. An increase in the amplitude of G–IG conditions was also characteristic for the post-MPT interval, which consequently led to thickening of ice sheets during glaciation periods and an increase in sea-surface and Antarctic interglacial temperature [108,109]. The demographic reconstruction performed with all chosen values of μ could explain the dramatic downturn of effective population size in the different seal species during that period (Figure 4, Appendix A). However, the broad range of the available μ assessments makes the association between bottlenecks and specific glaciation events ambiguous.

For the Baikal seal, the maximum decline encompassed up to 80% of population size, and is the most severe bottleneck observed among the studied seal species. It might indicate the differences in habitat among the seal species at that time. The advancing ice line may have forced the migration of ancestors of Baikal seals and driven them into rivers. During the Early Pliocene, the sea ice appeared at ~4.5 Mya at the level of Iceland [110], it became widespread during Northern Hemisphere glaciation at ~2.75 Mya [111], and after 1.5 Mya the ice sheet expanded repeatedly towards the continental shelf [112] along the western Barents Sea and Svalbard margins [113,114,115]. Based on μ = 1.2 × 10^−8^ and μ = 0.7 × 10^−8^, we could suppose that the migration took place between ~1.5–0.6 and ~3–1 Mya, respectively. This period corresponds to the time of ice expanding and conform the earlier studies [2,3]. According to the seismic structure of the northwestern margin, at least sixteen glacial advances occurred during the last 1 Myr [116]. In addition, a path between the Baltic Sea and White Sea, through Lake Ladoga, appeared and disappeared multiple times during this period [117,118]. At least, during the Eemian interglacial period (115–130 ka ago), all this area was covered by a single sea basin, desalinated due to river inflow [118,119]. It is an outstanding question of when isolated regions were completely covered or not by ice, but theoretically they could be inhabited by isolated populations of seals. If we take into account μ = 2.5 × 10^−8^, after the first massive peak at ~0.7 Mya, the decline in effective population size of the Baikal seal was reached between ~0.4–0.2 Mya. We could suggest that migration to Lake Baikal started at some time between ~0.7–0.3 Mya. A similar date (~0.4 Mya) for the migration to Lake Baikal was proposed earlier, using an assessment of mtDNA-based genetic distance between *Pusa* species [4]. Another significant event possibly related to the history of the Baikal seal, was when the Yenisey and Ob rivers were blocked by the Barents–Kara Ice Sheet, at approximately 0.08 Mya, and the West Siberian plain was covered by the West Siberian Glacial Lake at this time [120]. Reconstructions for all values of mutation rate showed stabilization of effective population size before ~0.1 Mya. (Figure 4). This might indicate that Baikal seals inhabited the lake since at least that period, and endured the Last Glacial Period (LGP), lasting from 115 ka until 11.7 ka [121,122,123].

### 4.5. Phylogenetic Analysis

The phylogeny of the true, or earless, seals (Phocidae) to date is considered to be mostly resolved, based on molecular and morphological data [3,32]. However, there are controversial relationships between some species, despite the abundance of genetic data. In particular, the placement of species among the *Phoca*–*Pusa*–*Halichoerus* lineages is not well resolved [32]. One of the reasons why it is a challenge to resolve the phylogeny of these species, could be due to the rapid radiation of northern ancestors over vast areas, and the formation of large populations with overlapping habitats. The speed of radiation, and thus speciation, could be confirmed by the phylogenetic tree we obtained, where divergence events are tightly spaced in time. According to our analysis, divergence between *Phoca* and *Pusa–Halichoerus* happened ~2 Mya (95% CI 2.6–1.4 Mya), and given that generation lengths of selected species are ~15–22 years, one can assume that sufficient effective generations have not passed for complete lineage sorting [32]. That might also be indicated by the high genome similarity of gray and Baikal seals, despite their different habitats, isolation from each other, and morphological differences [32]. Some researchers have even suggested merging all three genera into a single genus, *Phoca* [3]. The estimated divergence times are consistent with the previously estimated divergence within *Pusa*, and occurred within 1.7–0.8 Mya, based on Bayesian dating using 15 nuclear genes and near-complete mitochondrial genome sequences [3,31].

## 5. Conclusions

This work was to resolve the problem of the migration of the Baikal seal to Lake Baikal using the assembled genome. However, despite providing new insights, our study raises more questions. The important issue is the specific demographic trajectory of the Baikal seal. Potentially, it could be further elucidated through genome assembly followed by demographic reconstruction for the ringed seal (Arctic Ocean population) and Caspian seal. The Ladoga ringed seal and Saimaa ringed seal are less important, as they are known to descend very recently from the Baltic ringed seal [65]. In addition, whole genome sequencing of the entire Phoca–Pusa–Halichoerus lineage will resolve the remaining issues with phylogeny and dating of speciation events. Comparing the gray, harbor, spotted and Baikal seal genomes, we revealed several interesting megabase-scale inversions in chr6 in the Phoca lineage, which might be related to speciation. However, they might be just a feature of reference individuals, and only a population scale study will clarify this issue. This can be solved by sequencing 8–10 more samples of each species. Sequencing data of different populations can help to model demographic scenarios and gene flow among populations. At this stage, our assembly will provide a resource for a population genomic study of Baikal seals that will help inform ongoing conservation efforts for this species, and help to preserve its genetic diversity and understand the potential burden of mutational load.

## Figures and Tables

**Figure 1 genes-14-00619-f001:**
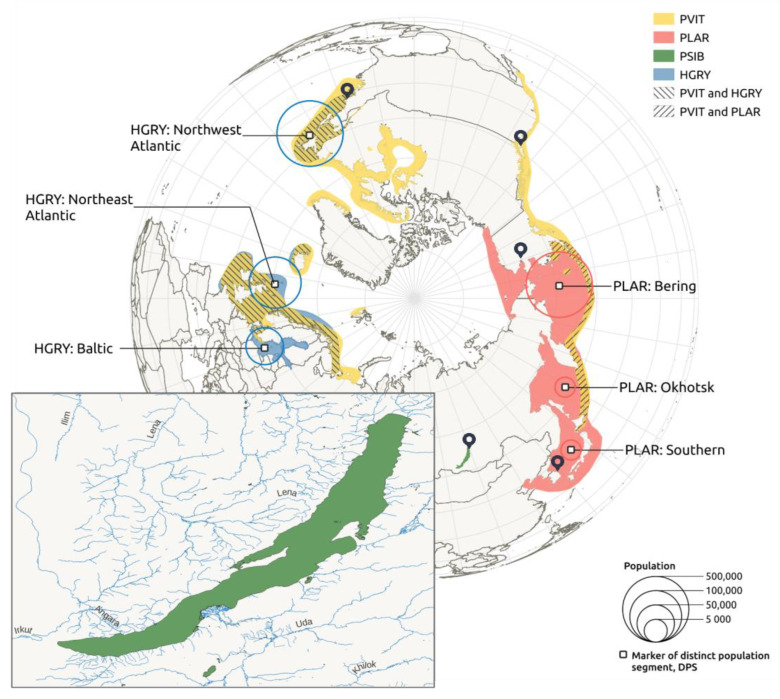
The geographic distributions of the four pinniped species included in this study: Baikal seal (PSIB), gray seal (HGRY), spotted seal (PLAR), and harbor seal (PVIT). Regions of habitat intersection are shown with diagonally shaded areas. Map pointers denote sample coordinates.

**Figure 2 genes-14-00619-f002:**
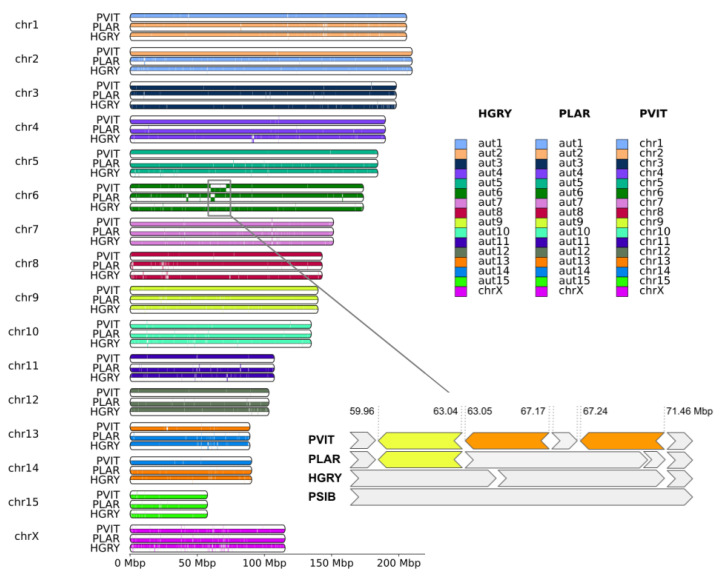
The synteny blocks between four pinniped species: PSIB was chosen as a reference and HGRY, PLAR, and PVIT were designated as the query species. Each chromosome was split into bottom and top parts: synteny blocks were drawn on the top part if it had the same orientation in both the reference (PSIB) and query species, and in the bottom part if the orientation was different. Chr 6 was enlarged, to emphasize the presence of three megabase-scale inversions. The yellow color on the enlarged pane highlights the single inversion common for PVIT and PLAR, and the orange color indicates the two inversions unique to PVIT. HGRY chr6 was reverse complemented in the small pane, and labeled with an asterisk on the small pane to avoid ambiguity.

**Figure 3 genes-14-00619-f003:**
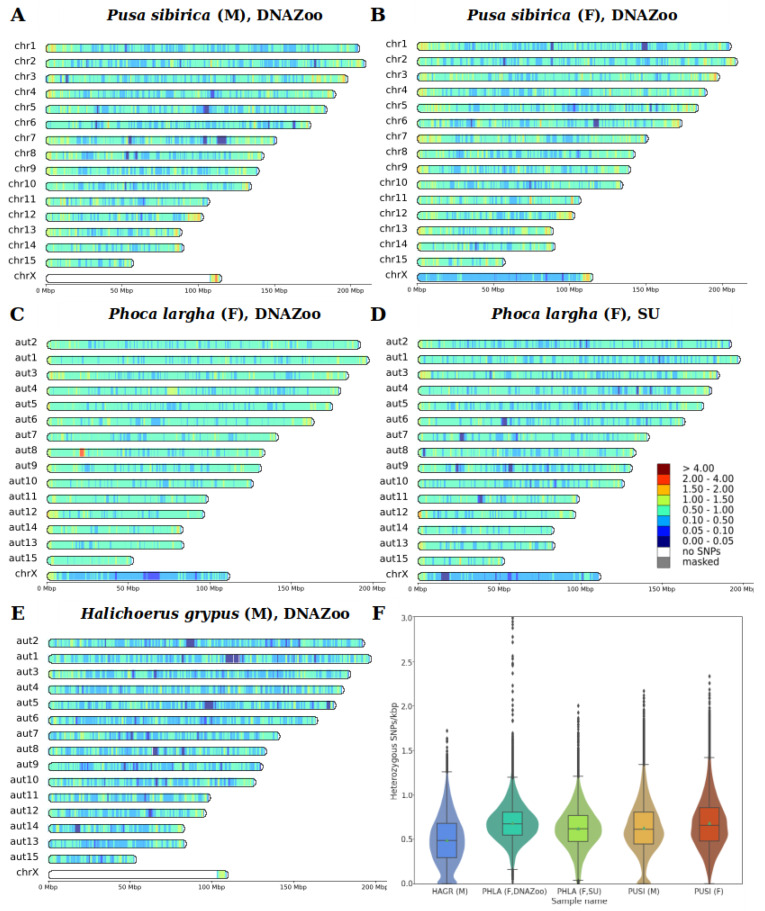
Heterozygous SNP densities (SNPs/kbp) for analyzed samples of three pinniped species. HetSNPs were counted in 1 Mbp sliding windows, with a 100 kbp step size, and scaled to SNPs/kbp. (**A**) PSIB, male, (**B**) PSIB, female, (**C**) HGRY, male, DNA Zoo, (**D**) PLAR, female, DNA Zoo, (**E**) PLAR, female, Seoul University, (**F**) violin and boxplots of heterozygosity distribution for all five pinniped samples. Autosomes of PLAR and HGRY were arranged according to their correspondence to PSIB and PVIT chromosomes.

**Figure 4 genes-14-00619-f004:**
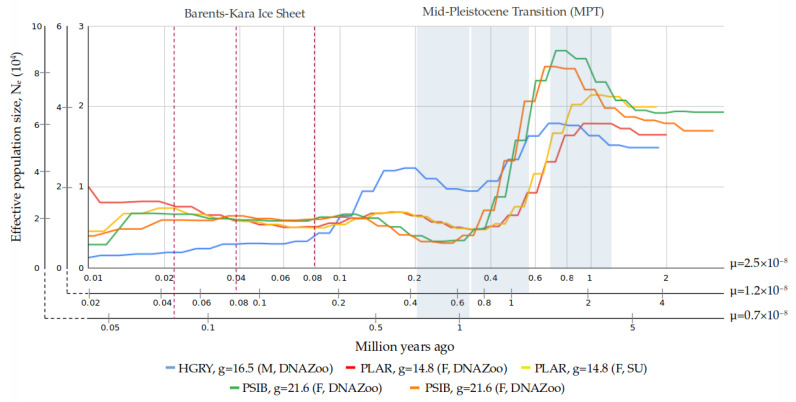
Demographic history reconstruction for all available samples of PSIB, HGRY, and PLAR. Three time scales were used for each value of mutation rate (μ = 2.5 × 10^−8^ substitutions per site, μ = 1.2 × 10^−8^, and μ = 0.7 × 10^−8^). Generation time (years) was set at different times between the species: PSIB—21.6 years, HGRY—16.5 years, PLAR—14.8 years.

**Figure 5 genes-14-00619-f005:**
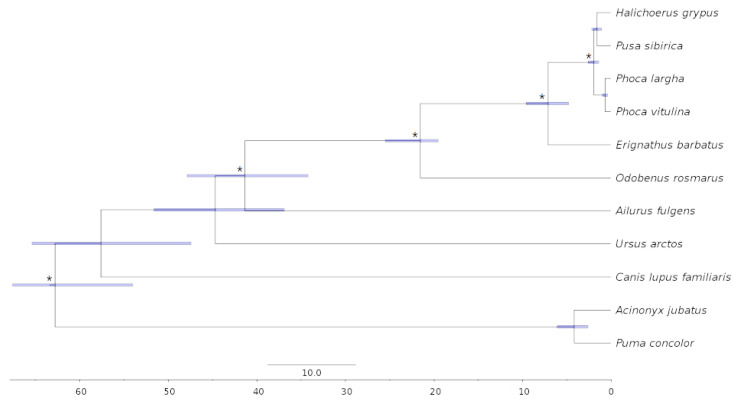
Dated phylogenetic tree of 11 carnivore species, including five phocid seals (HGRY, PSIB, PLAR, PVIT, and EBAR). Blue bars show the 95% confidence intervals. Asterisks (*) label nodes with fossil-based calibration priors (Appendix A). All nodes were supported at 100% based on 1000 bootstrap replicates.

**Figure 6 genes-14-00619-f006:**
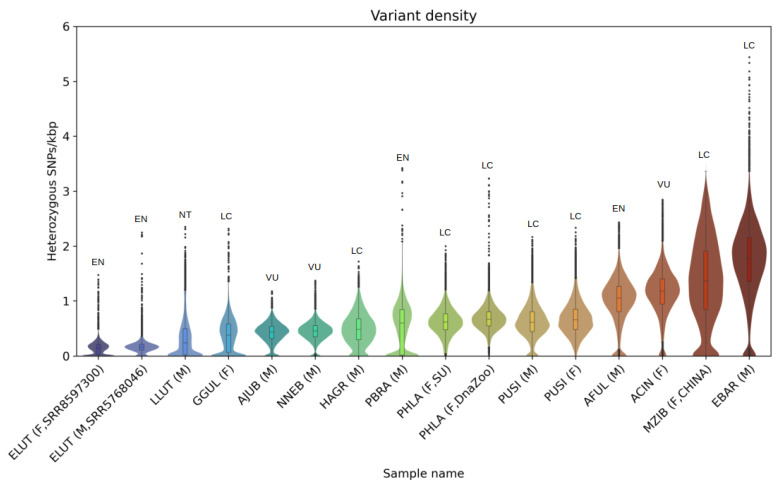
Violin and boxplots of heterozygosity distribution for analyzed samples of *Felidae, Ailuridae, Mustelidae*, and *Phocidae* species: *Enhydra lutris* (ELUT, EN), *Lutra lutra* (LLUT, NT), *Gulo gulo* (GGUL, LC), *Acinonyx jubatus* (AJUB, VU), *Neofelis nebulosa* (NNEB, VU), *Halichoerus grypus* (HAGR, LC), *Pteronura brasiliensis* (PBRA, EN), *Phoca largha* (PHLA, LC), *Pusa sibirica* (PUSI, LC), *Ailurus fulgens* (AFUL, EN), *Aonyx cinereus* (ACIN, VU), *Martes zibellina* (MZIB, LC), *Erignathus barbatus* (EBAR, LC). IUCN Red List categories: LC—least concern, NT—near threatened, VU—vulnerable, EN—endangered.

**Table 1 genes-14-00619-t001:** General description and quality metrics for genome assemblies of four pinniped species used in this study. MP = mate pair library, PE = paired-end library.

	Baikal Seal (PSIB)	Gray Seal (HGRY)	Spotted Seal (PLAR)	Harbor Seal (PVIT)
IUCN Red List Category ^1^	LC	LC	LC	LC
Gender	male	male	female	female
Assembly source	DNAzoo	DNAzoo	DNAzoo	DNAzoo
Assembly accession/reference	-	Halichoerus_grypus_HiC	Phoca_largha_HiC	GSC_HSeal_1.0_HiC
Sequencing/assembly approach	MP + HiC	Overlapping PE + HiC	Overlapping PE + HiC	Linked reads + HiC
2n	32	32	32	32
C-scaffolds	16	16	16	16
Scaffold N50, Mbp	151.5	141.6	142.1	152
Number of scaffolds	146,408	388,387	249,399	5311
Total genome length, Gbp	2.35	2.4	2.36	2.36
C-scaffolds length, Gbp	2.29	2.16	2.16	2.32

^1^ IUCN Red List category: LC—least concern.

**Table 2 genes-14-00619-t002:** Completeness of pinniped genome assemblies assessed using BUSCO v5; the mammalia_odb10 database of 9226 orthologs was used in the analysis.

Species/BUSCOs	Complete Single-Copy	Complete Duplicated	Fragmented	Missing
*P. sibirica*	7981 (86.5%)	140 (1.5%)	444 (4.8%)	661 (7.2%)
*H. grypus*	8352 (90.5%)	190 (2.1%)	242 (2.6%)	442 (4.8%)
*Ph. largha*	8434 (91.4%)	178 (1.9%)	215 (2.3%)	399 (4.4%)
*Ph. vitulina*	8651 (93.8%)	198 (2.1%)	90 (1.0%)	287 (3.1%)

**Table 3 genes-14-00619-t003:** Statistical metrics of heterozygosity for analyzed samples of three pinniped species. SNPs are counted in 1 Mbp sliding windows, with a 100 kbp step size, and scaled to SNPs per kbp.

Species	Min	Median	Mean	Mode	Max
*H. grypus, male*	0	0.48	0.49	0.48 ^1^	1.72
*Ph. largha*, female (DNAZoo)	0	0.67	0.68	0.66	3.23
*Ph. largha*, female (Seoul University)	0	0.62	0.62	0.60	2
*P. sibirica*, male	0	0.61	0.63	0.56 ^1^	2.17
*P. sibirica*, female	0	0.66	0.68	0.56	2.34

^1^ For male samples we observed an additional mode at 0. It corresponds to the hemizygous region (the part of the X chromosome outside of the PAR) in males.

## Data Availability

De novo assembly of male Baikal seal and related reads are available from BioProject PRJNA905417. Hi-C data is available from DNA Zoo SRA BioProject PRJNA512907. Corresponding SRA accessions are listed in Appendix A. Resequencing data for female seal are available from BioProject PRJNA905443 under SRR22409450 accession. Interactive Hi-C contact maps for the Baikal seal are available at dnazoo.org/assemblies/Pusa_sibirica.

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
