# Peer review of "Chromosome-Length Assembly of the Baikal Seal (Pusa sibirica) Genome Reveals a Historically Large Population Prior to Isolation in Lake Baikal"

_genes, 2023, doi:10.3390/genes14030619_

Round 1

Reviewer 1 Report

This manuscript presents a high quality, chromosome-level assembly of the Baikal seal. This species is interesting for its isolation in Lake Baikal, making it the only pinniped species that lives exclusively in fresh water. As the authors point out, the question of how and when the Baikal seal arrived at Lake Baikal is an important biogeographic question that drives their research. Through their high-quality assembly, they are able to reconstruct a dated phylogeny of some phocid seal species, assess patterns of heterozygosity across the genome of the Baikal seal and other seals, and infer the past demography of the Baikal seal. As they discuss, these interesting findings still cannot fully address questions about the evolutionary history of the Baikal seal. Still, this new reference assembly presents an important step forward in future genomic studies of this and other seal species.

Overall, I find the genome assembly to be very high quality. It follows the published pipeline established by the DNA Zoo Consortium (co-authors on this paper), and the quality assessments performed here show that it is a high-quality assembly. The comparative karyotype work shows further connection to the chromosome-level quality of the assembly.

I have only one major issue with the manuscript: the justification for the chosen mutation rate. I outline my concerns and suggestions below, followed by additional minor comments listed by line.

It is hard to see how 2.5x10^-8 is chosen as the most appropriate mutation rate for this species. The authors cite Hoffman et al. 2022 for the 2.5x10^-8 mutation rate, but Hoffman et al. cite Dornburg et al. 2012, which is a paper on Cetacea. It is not clear why the authors choose to not use the estimate from Peart et al. 2022, which directly estimated a per-generation mutation rate of 0.7x10^-8 across phocid species. This 3.5-fold difference in mutation rate clearly makes a significant difference in the inference of timing of past demographic events, as they show in Figure S4. The authors appear to recognize this issue in the discussion and at first state “However, the broad range of the available μ assessments makes the association between bottlenecks and specific glaciation events ambiguous.” In the following paragraph, however, they rely heavily on the 2.5x10^-8 mutation rate to align demographic changes with specific glaciation events. I would strongly encourage the authors to consider using the pinniped-specific 0.7x10^-8 mutation rate for their demographic analyses. If they decide against that, I think that decision should be justified in the methods or discussion. At the very least, I think that the results from the 0.7x10^-8 mutation rate should be discussed in the context of geological events as well.

Minor comments:

L53: Add “(Pusa sibirica)” after the first mention of the Baikal seal in the introduction.

L62-64: Please be more explicit about where this discrepancy in estimates comes from, and how it relates to the methods used, especially timing based on “genetic distance between related seal species”

L90-91: I think it is worth reporting the traditional population estimates of this species in number of estimated individuals, in order to compare later to the genetic estimates.

L266: Add “respectively” after “4.92%.”

L337: Add “seal” after “spotted” (or alternatively, change this to “Gray, spotted, and Baikal seals”)

L453-455: I have a hard time seeing this direct connection. A uniform distribution of heterozygous sites likely mean low inbreeding and a population near equilibrium. If you think it specifically supports the demographic scenario you propose in this sentence, you should explain more clearly the connection.

L532: Are the authors implying their results support or do not support merging all species to Phoca? Please be more explicit here.

Figure 6: Since IUCN status seems to be important here, can you indicate it on the figure itself (by color or symbol) rather than just in the description?

Reviewer 2 Report

The study creates the first genome de novo assembly of a rare pinniped freshwater species to investigate heterozygosity and demographic history. The manuscript although relatively basic in information provides some baseline data for more comprehensive studies in this species and to conduct comparisons with closely related species.

The manuscript has a significant number of English language errors that should be corrected by a native English speaker, some of which are detailed below.

The discussion is relatively well written and overall there is a great deal of data using only 2 individuals. I would suggest the authors detail the importance of this study for the future and how more detailed field sampling and genomic sequencing may reveal important insights into demographic histories that may allow greater inference of how the species became diverged and the role of genetic drift and other environmental pressures on contemporary genetic structure.

The final paragraph of the introduction is the most important to set up the study, however I find it lacking in detail and substance to justify this study. There should be more detailed discussion of the knowledge of genetic diversity in this species or closely related species and how demographic histories may shape these species genetically also under hunting pressure which may also interfere with the demographic histories of an isolated species.

The methods are fairly detailed for replication and there are only minor typos and improvements recommended.

Minor corrections

L 48 inconsistent use of kya or Mya please choose one.

 L53 suggest revising 1st sentence saying something is a. mystery is very vague. In what sense is it a mystery would be more informative to the reader.

State latin name of Baikal seal here

It would be nice to see Fig 5 first and a reference to this early on to let reader become more informed about the study area. Adding in the river connection would also be very informative but just a suggestion, Zoomed in map of the Lake Baikal and the nature reserves/protected areas would help a great deal.

L63 what are the genetic distance estimates from citation

L64 citation

L73 suggest “on rocks and carvings out of bone and stone more than 7,000 years ago”

L78 suggest greater instead of as intense

I'm not sure what you mean protect commercially important fish. Do you mean legislation began protecting commercially important fish?

L77 what caused poaching protection i.e what population census size did this species reach this would then relate to demographic analyses I am expecting later on

L82 Contrary to popular beliefs, although suggest rewording the sentence.

L86 which are swallowed whole.

L89 why targeting molted pups

L95 consider changing danger to predators

L98 I don’t understand this sentence please rephrase

L103 reword- Genetic diversity is a result of demographic history and selective events that may have been shaped by environmental conditions, gene flow and genetic drift etc.

L108 what additional data

Phylogenetic trees

Inference of past events not understanding

Remove and L112

I still don’t like mystery please rephrase.

L127 what size bp was you library reads

L141 why did you downsample

L162 why the stone marten??

L174 which reference genome

L239 what version figtree

Table 1 very difficult to read the 1st column and how it is broken up please improve presentation

Table 2 keep the order the same as table 1 for each species

L281 suggest exact is not correct wording here

Fig 3 top of Barents cut off top off figure Y axis add Ne in parentheses

L383 long read sequencing.

L477 results should not be in discussion.

L502 rephrase outstanding question – suggest important

L508 use different citation method to journal requirements for Sasaki et al

L537 remove 1st sentence and rephrase

L539 remove of course, main suggest- important

I would suggest by increasing sample size to estimate population level heterozygosity and diversity. Investigation of diversity within the species range
